# The Assessment of the Quality of Human Resources in the Midwife Profession in the Healthcare Sector of the Czech Republic

**Erika Urbánková * , Petra Hospodková and Lucie Severová**

Department of Economic Theories, Faculty of Economics and Management, Czech University of Life Sciences, Prague 16500, Czech Republic; hospodkova@pef.czu.cz (P.H.); severova@pef.czu.cz (L.S.)

* Correspondence: urbankovae@pef.czu.cz; Tel.: +420-224-382-230

**Abstract:** The objective of this paper is the assessment of the current state of employment in the midwife profession in the health and social care sector, especially from the viewpoint of quality assurance. Primary research focuses on the level of practical and theoretical knowledge and the skills of graduates within the last 10 years, as well as the forms of development of human resources that healthcare facilities offer for the purposes of supplementing knowledge and developing skills that are lacking. The quantitative research took place in private and public healthcare facilities in the Czech Republic in 2017. The results of the quantitative research show that the offers of workers in the midwife profession in the labor market are slightly insufficient in relation to demand. Research results show that the overall educational level of graduates has improved over time. The research also shows that the greatest deficiencies in terms of missing competencies among new graduates are seen in the area of expertise according to Regulation No. 55/2011 Coll. (newly No. 2/2016), and that what the graduates lack most are skills in communicating with patients. Conversely, current graduates are better equipped with language skills, computer skills and time-management ability. The research also shows that overall practical readiness lags far behind theoretical readiness.

**Keywords:** Czech Republic; healthcare; midwife; human resources quality and development; graduates; cluster analysis

**JEL Classification:** I11; J; O15

## 1. Introduction

Human capital is of great importance in the modern economy (Hrabinová et al. 2012). Professional competence specified by Act No. 96/2004 Coll. (amendment 201/2017) is required for the performance of the midwife profession. The activities of midwives are stipulated in Sec. 5 of Regulation No. 55/2011 Coll. (or newly No. 2/2016), on the activities of healthcare workers and other professionals. Whereas in previous years professional competence was contingent upon completion of study at a secondary medical school with nursing for women or midwifery as the main subjects, or graduation as a certified midwife from a three-year study program at a higher medical school, at present, professional competence results from the completion of at least a three-year accredited medical bachelor's degree for future midwives. Currently, the Czech Republic provides education for future midwives in bachelor degree study programs at 11 universities. From the viewpoint of classification according to the International Standard Classification of Occupations (ISCO) at the 5th level, this profession has been included in the Midwives with Certification (ISCO 32221) category. However, with regard to the legislative regulation resulting from Act No. 201/2017 Coll., it is also possible to consider ISCO 32222, i.e., Midwives without Certification (Portál veřejné správy 2017).

A fundamental change came into force on 1 September 2017, whereby the authorization to pursue a medical profession without professional supervision is no longer contingent upon obtaining a certificate for the performance of a medical profession without professional supervision. According to law, midwives in the Czech Republic, as in other EU countries, can work independently, i.e., without professional supervision and without a physician's referral in the case of physiological pregnancy, childbirth, and puerperium. After the Czech Republic's accession to the EU, Czech midwives can work under comparable conditions to those of any of the member countries. Similarly, midwives from other EU countries can work in the Czech Republic.

According to data published by the National Institute for Education, from the viewpoint of the educational structure of this profession, until 2015 the majority of professionals had only secondary education. That represents approximately 81% of all persons employed as midwives. The remainder consisted of people with tertiary education, of which 55% had a bachelor's degree. Due to the current midwifery education system, the share of those with a higher specialized bachelor's or master's degree is growing. In 2015, 19% of nurses and midwives achieved this level of education, while in 2012 it was 14% (Chodounská 2016). The objective of our primary research is the assessment of the current state of employment of midwives in the health and social care sector, especially from the viewpoint of quality assurance. Primary research focuses on the level of practical and theoretical knowledge and the skills of graduates within the last 10 years, as well as on the forms of human resources development that healthcare facilities offer for the purposes of developing lacking skills and supplementing knowledge.

Monitoring and evaluating of human resources in healthcare is a key activity for setting up an ongoing learning platform. As has been said in the last decade, there have been major legislative changes in the Czech Republic in the field of nursing, which directly reflect the setting of requirements for midwifery teaching. A partial objective of this study is to identify the competencies of midwives which, in practice, appear to be inadequate with regard to the setting up of the education system. Education as a factor that directly promotes the quality of health care was analyzed in 2013, and the results speak for lifelong learning in nursing (Beňadiková 2018; Vacková 2018). An inventory of the most important characteristics of the different midwifery systems is summarized in a study by Emons and Luiten (2018). This survey therefore focuses on the job responsibilities and competencies of midwives, their position within the health care system, training and statistics, such as the numbers of midwives and their income. The next study promotes discussion regarding the mandated requirements for allocated clinical practicum hours and specified numbers of clinical-based skills (Ebert et al. 2018). Supervision models to develop and implement a best practice model in midwifery education programs are also currently analyzed (Mckellar and Graham 2018). The researches also supported the idea that the major learning impact for students is related to the midwife they worked with each day, and highlights the importance of practice (Licqurish and Seibold 2018). Many contemporary authors also deal with the need of a new competency framework, and aimed at implementing a more standardized and evidence-based method to learn and assess competencies, as well as to guide continuous competency development in practice (Embo and Valcke 2018). There is currently no comprehensive survey in the Czech Republic that retrospectively analyzes the soft and hard competencies of midwives over time in relation to the learning platform.

The creation of the concept of the development of human resources signifies that people are not only the workforce of the market, but they are also one of its most important organizational resources. The concept of human capital has appeared in the last few decades. The economists of the Chicago School, T. W. Schultz, G.S. Becker and J. Mincer, are the founders of the theory of human capital. They invented the theory of human capital in the early sixties of the 20th century. The qualitative component of human capital can be observed by the number of employed and unemployed according to their level of education, by the numbers employed in different professions (in addition to the level of education, this number reflects informal education, skills, and experience), and by the average length of education (Becker 1964). Unsuitable education (or insufficient education) represents a structural problem for the economy on the labor market. Structural unemployment

represents an incongruity on the labor market, which comes about primarily as a result of structural changes in the economy, whereby certain sectors (professions) are expanding and others are fading away or declining. Under structural inequality, we cannot assign a suitable worker to a job because they have inappropriate qualifications. For the reduction of structural unemployment, continuous education (training, requalification) of the labor force and growth of its spatial mobility are key (Samuelson and Nordhaus 2008). However, the process of pairing in the labor market is hampered by regional differences. Regional (spatial) disparities express the extent of the difference in the analyzed economic phenomenon seen within the scope of the regions of a given country. According to Hučka, "We understand disparity to be every difference or inequality, whose identification and comparison has a certain purpose (social, economic, political, etc.). We then understand regional disparity to be a difference or disproportion in various phenomena or processes having a definite territorial location (it can be allocated within a defined territorial structure) and occurring within at least two entities of such a territorial structure)" (Hučka 2007, p. 14). The perspective that is used by most authors in research is the material one, according to which most authors lean toward the classification of disparities as economic, social, and territorial. Disparities are classified from a horizontal perspective into material and immaterial disparities, whereby both include the economic, social and territorial spheres of occurrence. Further, disparities are classified according to a vertical perspective into global-level, European-level, national-level, and local-level (Vorauer 1997).

## 2. The Models and Methods

The aim of this paper is to assess the current state of employment of midwives from the viewpoint of the health and social care sector based on the conducted primary research. A method of quantitative research was used, namely, a questionnaire survey.

The research focused primarily on the issue of quality assurance. The level of the practical and theoretical knowledge and the skills of graduates from the relevant medical field of study in the last 10 years were assessed, as well as the form of development of human resources and the area of non-formal education. The questionnaire survey contained closed questions and used score points.

The quantitative research and data collection took place in private and public healthcare facilities in the Czech Republic in 2017 over a five-month time period. The questionnaires were distributed in three rounds. Ninety-five healthcare facilities were contacted; these include neonatal departments, or have sections that follow the departments of the postpartum gynecological-obstetric department. This was the total number of healthcare facilities for 2017 (Aperio 2017). The questionnaires were sent to nurses and chief nurses (at least 10 years of experience in this position was required) in departments of gynecology and obstetrics throughout the Czech Republic. Seventy-one questionnaires were returned (after the third completed round for all three rounds); hence the questionnaire return rate was 75%, of which 68 questionnaires may be deemed relevant (3 questionnaires had not been completed). The questionnaire contained ten closed questions, three of which were identifying. The questions were focused on the quality and quantity of the midwife profession. The questionnaires were distributed in electronic form. In selected healthcare facilities in the Central Bohemia region, the quantitative questionnaire survey was also accompanied by structured interviews with managers.

Graphical representation and the descriptive method were used to evaluate the questionnaires. Furthermore, a multidimensional statistical method of cluster analysis, serving to find homogeneous regions and identifying differences among the studied regions, was used for the paper. Groups of homogeneous objects were grouped into clusters, so that objects belonging to different clusters were mutually heterogeneous. The statistical program STATISTICA 12 was used for the calculations. The hierarchical clustering process is captured in a special tree graph (dendrogram) which illustrates the metric distances of the cases. Ward's method, which is based on creating clusters with the highest possible internal homogeneity, was used for hierarchical clustering, and it is specific by requiring the distance of objects to be expressed as a square Euclidean distance (Hindls et al. 2007; Meloun et al. 2011).

Ward's method:

$$\Delta C = \sum_{i=1}^{G} \sum_{j=1}^{n} \left(x_{gij} - v_{gi}\right)^2 - \sum_{i=1}^{A} \sum_{j=1}^{n} \left(x_{aij} - v_{aj}\right)^2 - \sum_{i=1}^{B} \sum_{j=1}^{n} \left(x_{bij} - v_{bj}\right)^2 \qquad (1)$$

where: $x_{gij}$ is the value of the *i*-th element of the cluster *G*, *G* is the number of elements in this cluster, $v_{gj}$ is the average value of the *j*-th cluster variable *G*, etc.

The metrics of squared Euclidean distances:

$$D_E(x, y) = \sum_{i=1}^{n} (x_i - y_i)^2 \qquad (2)$$

where: *x* is the current value of the component (variable) within the matrix and *y* represents the number of factors.

In the event that the input database contains macroeconomic variables in various units of measure, and the factor analysis method is not utilized prior to the cluster analysis itself, and the data are only reduced with the aid of a pair correlation matrices, it is necessary to standardize such data before entering them into the cluster analysis. The most common method used for the purposes of the analytical portion of the work is standardization by way of the *z*-score (Hebák et al. 2007):

$$z_{ij} = \frac{x_{ij} - \mu_i}{\sigma_i} \qquad (3)$$

where: *x* is the current value of the element in the matrix, $\mu$ is the mean value and $\sigma$ is the standard deviation.

## 3. Data

For a long time, the midwife profession has lacking human resources in the labor market. According to the Ministry of Labor and Social Affairs, fluctuation of workers in this professional group is slightly above the average, and this profession belongs among groups with slightly higher mobility (Further Education Fund 2017).

The conducted primary research is fully supportive of this statement, as 17 of the addressed healthcare facilities stated that they are understaffed, and another 40 that they are slightly understaffed (see Figure 1). To the question: "Based on your experience, assess the current number of employees in the given position", none of the facilities responded that they were overstaffed in this profession. Slight overstaffing was reported by only one facility located in Central Bohemia, and the ideal state was reported by ten facilities in the Czech Republic. To the question: "Based on your experience, estimate the future filling rate for the given position", for the mid-term horizon of 5 to 10 years, none of the facilities responded that it expects to be overstaffed or to be slightly overstaffed in the future. Ten percent of the respondents predicted ideal staffing, 55% predicted slight understaffing, which is the most frequent answer, and 35% of the respondents were inclined to expect understaffing; hence, they assume that they will have to face a lack of staff in this profession. These are the estimates of managers—ward or chief nurses—who have been working in the relevant positions for at least ten years, and who hold pessimistic expectations based on practical experience and long-term problems with filling the capacities and vacancies in quantitative terms.

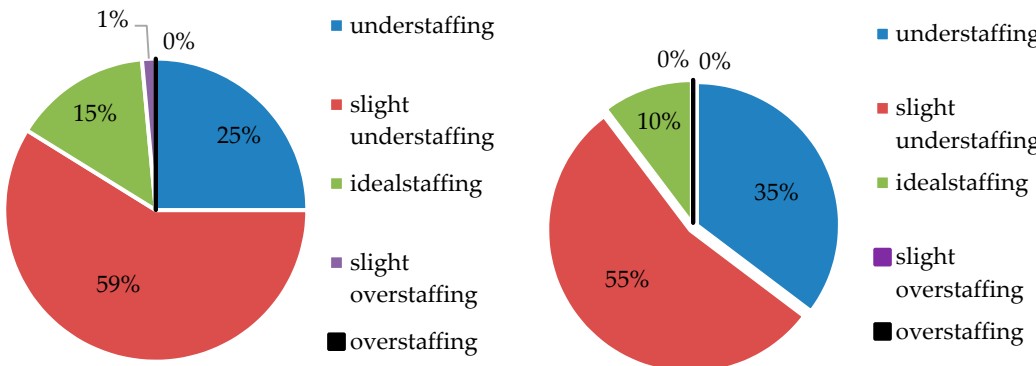

**Figure 1.** Filling of the vacancies in the midwife profession (**left**) and Estimation of the filling of future vacancies in the midwife profession (**right**).

Furthermore, respondents were asked: "How do you evaluate the theoretical and practical readiness of today's graduates to perform the midwife profession", which inquired into the level of theoretical and practical knowledge necessary for the performance of the midwife profession (Figure 2). The most frequent answer was an average level of readiness. There is the predominant opinion that current graduates are rather better prepared for the performance of the given profession.

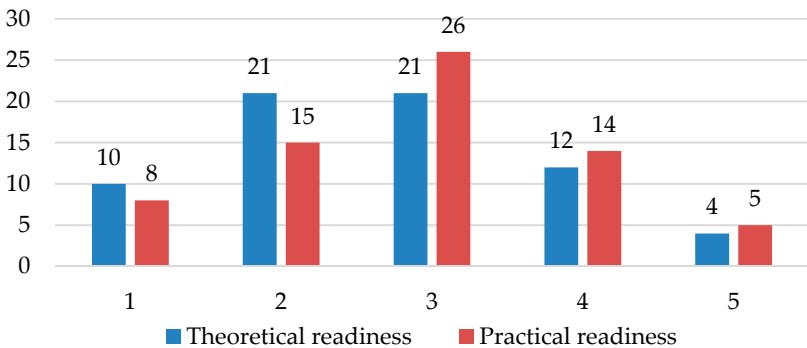

**Figure 2.** Theoretical and practical readiness of current graduates for the midwife profession (1 = excellent, 5 = insufficient).

Overall, however, the research shows that practical readiness lags behind theoretical readiness. An excellent level of graduates' theoretical and practical knowledge was observed by only 18 healthcare facilities. There were more facilities with excellent theoretical readiness (10 facilities, 15% of respondents) than there were facilities with excellent practical readiness (8 facilities, 12% of respondents). The facilities rate theoretical readiness as excellent, especially in Prague, Central Bohemia, and Moravia-Silesia. Practical readiness is rated as excellent by one facility in each region. The theoretical readiness of graduates is rated as insufficient, especially in Central Moravia, the Northwest, and the Southeast of the Republic. The practical readiness of graduates is rated as insufficient, especially in the Southwest, Northwest, and Southeast of the Republic. Average theoretical readiness was reported by 31% of the respondents and 38% of the facilities reported average practical readiness. Very good theoretical readiness was reported by 31% of the respondents, while 22% reported average practical readiness.

From the viewpoint of competencies (knowledge, abilities, and skills) for the performance of the given profession, the investigation focused on the extent to which the newly-starting graduates lack individual professional competencies. For the purposes of the survey, 6 basic competencies deemed as relevant for the midwife profession according to the Occupational Information Network were selected (O\*NET Online, 2017).

The first area of knowledge to be evaluated was language competence, which is particularly important with regard to growing multicultural demands. Further, the ability to work effectively within a team, including conflict management, teamwork, problem solving, social responsiveness, etc., was assessed. The third assessed area was the level of expertise necessary to perform the scope of activities specified in Regulation No. 55/2011 Coll. (newly No. 2/2016). Other assessed areas were computer skills, skills in communicating with patients, and the ability to organize work in terms of time-management.

To the question (Figure 3) "Sorted by frequency, what skills are the graduates lacking in the relevant position?", the majority of healthcare facilities responded that current graduates lack most professional knowledge according to Regulation No. 55/2011 Coll. (newly No. 2/2016) (especially in the Southwest and the Northeast regions), as well as skills for communicating with patients (especially in the Southeast and Central Bohemia regions). Conversely, current graduates are best equipped with language skills, computer skills, and the ability to organize their work.

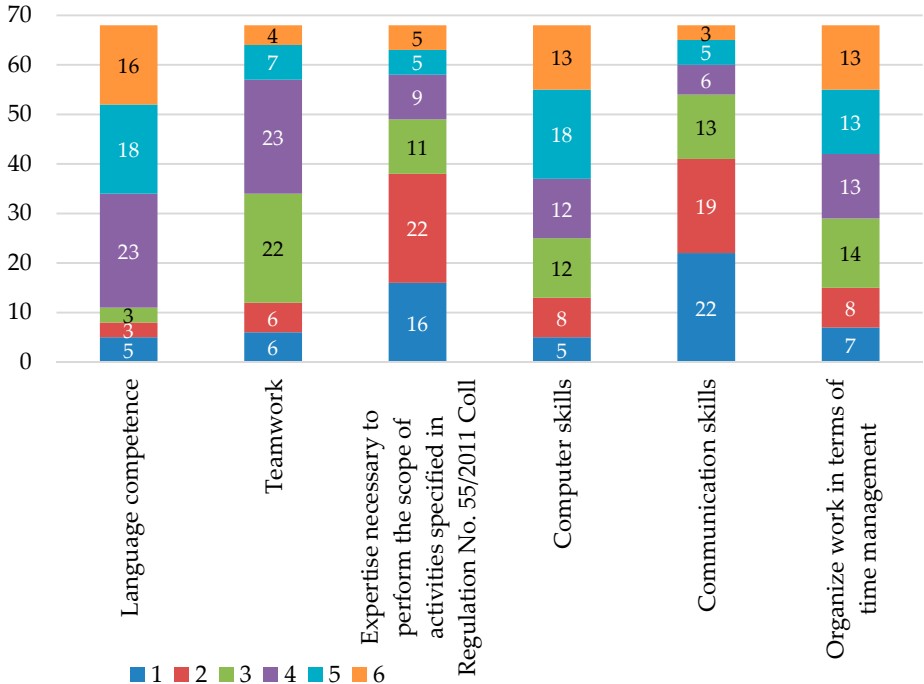

**Figure 3.** Competencies in new graduates (1 = lacking the most, 6 = lacking the least).

Changes in educational structure are typical for the midwife profession. Definitely the most fundamental step was the alignment of the education system with European requirements, i.e., EU 80/154 EEC on the requirements for the training of midwives. The objective of the research was to establish how the knowledge level of graduates changes over time, and whether changes in the education system are reflected in the level of the graduates' education.

The issue of workforce qualifications being discordant with the qualification requirements of vacancies is linked in economic theory to the concept of structural unemployment. Structural unemployment arises if what is offered by workers in a particular profession (field of study, level of education, professional expertise) does not correspond to the demands of the employers in the same profession. Hence, quantitative discordance is observed; however, qualitative issues may also bring about a situation in which a graduate from a given field of study qualitatively fails to meet the requirements of the relevant position in practice (Kaufman and Hotchkiss 2002; Layrd et al. 2005).

Plesník (2007) notes that the continuous training of the workforce (based on the idea of lifelong learning) and an increase in the spatial mobility of the workforce are the keys to

the reduction of structural unemployment. Formal education relates to the school education system and distinguishes between primary, secondary and tertiary levels of achieved education. Non-formal education is represented by a professional training process when the individual is already working. Trhlíková, Úlovcová, Vojtěch, and also Cazes and Nešporová (Trhlíková et al. 2006) highlight the fact that the fields of study offered by school institutions are designed to conform to students' requirements, regardless of the future ability of graduates to work in the given field on the labor market, and very often in practice do not correspond with the real demands.

To the question (Figure 4), "How do you rate the change in the overall level of knowledge of graduates in the midwife profession over the last 10 years?", only 27% of respondents stated that the graduates' knowledge level has stayed approximately the same over this time. The predominant opinion (41%) is that this knowledge level has improved over the last 10 years, and 10% of the respondents even deem that the improvement is significant.

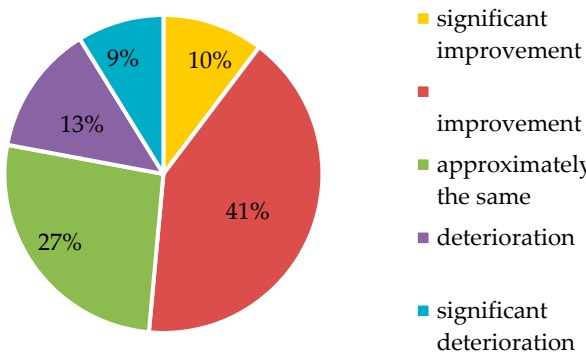

**Figure 4.** Overall development of the knowledge level of graduates in the midwife profession over the past 10 years.

The area of education is very important for healthcare professionals, which is also evidenced by the obligation to participate in lifelong learning in medical professions other than that of a doctor of medicine. According to Act No. 96/2004 § 53, lifelong learning is defined as the "continuous renewal, improvement, deepening and acquisition of knowledge, skills and competencies by healthcare professionals and other professionals in the relevant field, in line with the development of the field and the latest scientific knowledge, to maintain safe and effective performance of the profession concerned" (Portál veřejné správy 2016).

However, as of 1 September 2017, the amendment to Act No. 96/2004 Coll (amendment 201/2017) ended the registration, linked to lifelong learning in the form of a credit system, of medical professions other than doctors of medicine at the National Centre of Nursing and Medical Professions Other than the Profession of a Doctor of Medicine (NCONZO) (Portál veřejné správy 2016). The result of this change is the abolition of education through the credit system and the fact that work without professional supervision is no longer subject to registration with the NCONZO. Lifelong learning is still mandatory, but currently there is no platform that precisely defines a system of lifelong learning controls. The survey has discovered that 59% of the respondents agreed with the cancellation of the credit system, while 41% did not agree with this change. Thus, the survey shows that this is a very controversial issue that polarizes the professional public.

The survey was also intended to establish what educational activities are currently pursued the most in the midwife profession. In particular, the share of internal and external educational activities in the workplace was to be established. The question was: "Sort the educational activities from the offer depending on the extent to which they are used by midwives in your department". The results (Figure 5) show that, despite the abolition of the credit system, the focus is on external educational activities, with the largest share belonging to external specialized seminars and educational events (28 healthcare facilities, i.e., 41% of the respondents). The results of the survey also show that the healthcare facilities

use external conferences and congresses, whose share is approximately the same as the share of internal specialized seminars and educational events. The possibility of pursuing a master's degree study program while working is the lowest. The opportunities for the funding of lifelong learning are limited within the departments concerned; therefore, the relevant profession often pays for the education from its own resources. Respondents were asked whether the continuous education of an employee working as a midwife is reflected in the flexible component of the salary or wages. The answer was clearly positive in only 3 cases, while 47 healthcare facilities indicated that the willingness to learn is only partly reflected in the salary or wage, and 18 healthcare facilities responded that the employee's willingness to learn continually has no effect on the flexible component of their salary or wage.

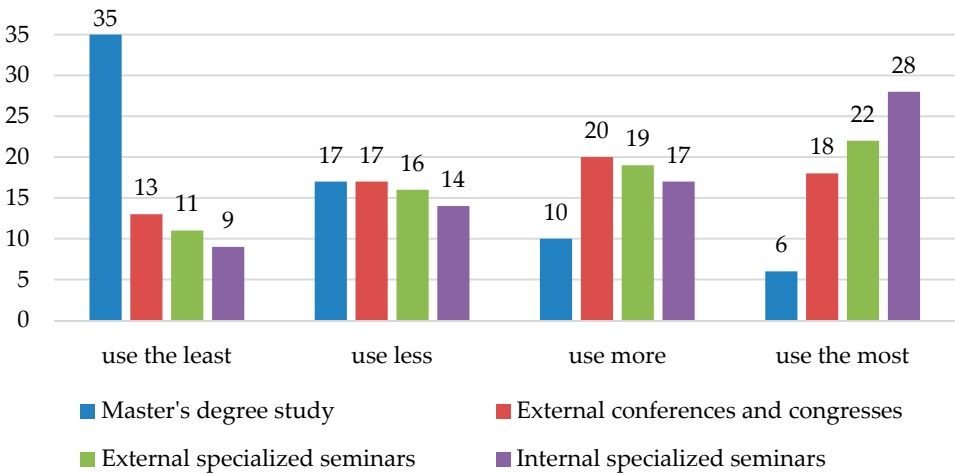

**Figure 5.** Forms of education currently used in departments of gynecology and obstetrics.

## 4. The Results

The cluster analysis serves primarily to find homogeneous regions and identifies the differences among the observed regions. Homogeneous objects are grouped into clusters, such that objects belonging to different clusters are heterogeneous with each other, and the K-means method identifies the differences among the regions studied. Primary data entered for cluster analysis were modified to eliminate the impact of the number of healthcare facilities in the individual regions, and to maintain the possibility of interregional comparison. Healthcare facilities in the Czech Republic were differentiated according to their area of operation, for which the NUTS2 division of the regions was used (Figure 6). NUTS2 includes eight regions: Prague (PHA), Central Bohemia (SC), the Southwest (JZ), the Northwest (SZ), the Northeast (SV), the Southeast (JV), Central Moravia (SM), Moravia-Silesia (MS).

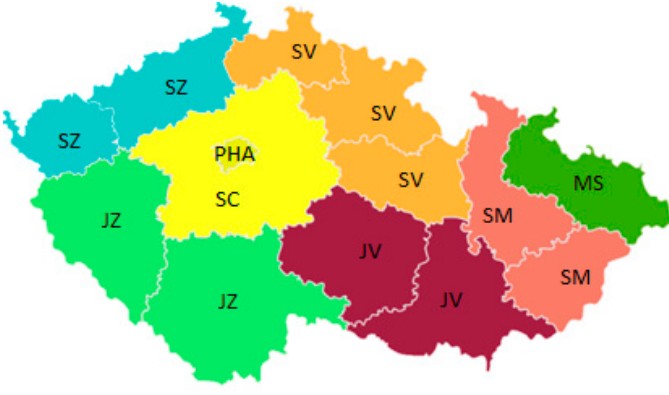

**Figure 6.** Czech Republic divided according to NUTS2.

Using the respondents' experience, the healthcare facilities evaluated the current number of employees in the relevant position, and also estimated the future staffing of the relevant position. The current situation is as follows: none of the regions reported overstaffing, while only one facility in Central Bohemia reported slight overstaffing. The most frequent answer was slight understaffing (40 responses) followed by understaffing (17 responses), and then the ideal status (10 responses). As regards future estimates, none of the facilities reported overstaffing or slight overstaffing; the dominant response was slight understaffing (37 responses), followed by the ideal status (7 responses). It is clear that the angularity of the frequency of the responses leads to pessimistic expectations, and that the expectations of understaffing are increasing.

Figure 7 shows homogeneous clusters of regions for the current capacity assessment (left) and the foreseen future capacity (right). As regards assessment of the filling of the current capacity, it is clear that two basic mutually very heterogeneous clusters have been formed. The first cluster is represented by Prague, Central Bohemia, Moravia-Silesia, and the Southwest, and using the K-means method, these regions have a higher rate of responses stating understaffing than other regions. The second cluster is represented by the Northwest, the Northeast, the Southeast, and Central Moravia, and using the K-means method, these regions show higher rates of responses stating slight understaffing and the ideal state. In the framework of the future capacity assessment, two basic clusters have again emerged; however, they have different regional representation. The first cluster consists of Prague, the Southwest, and Moravia-Silesia, and these regions again show slightly higher levels of concern, with respect to understaffing, than other regions. The second cluster brings together Central Bohemia, North Moravia, the Northwest, the Northeast, and the Southeast of the Republic, and is dominated by stronger concerns with respect to slight understaffing.

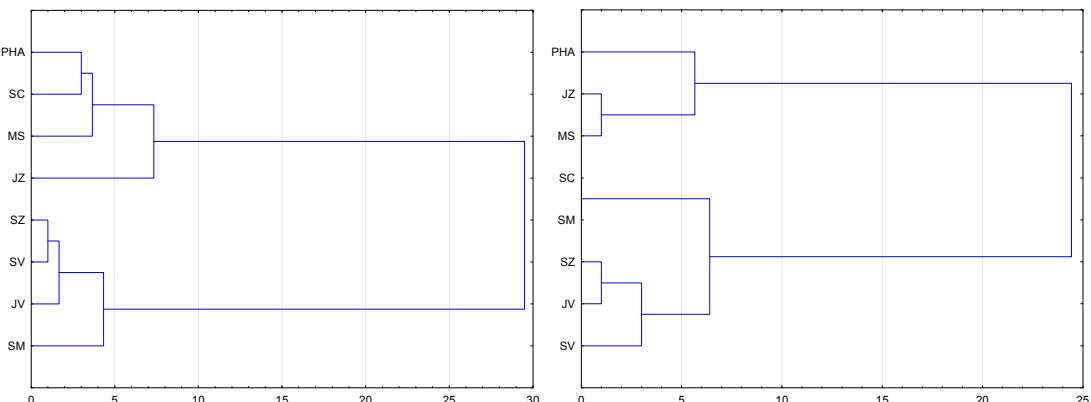

**Figure 7.** Hierarchical clustering, current capacity assessment (**left**) and foreseen future capacity (**right**) of the midwife profession.

Figure 8 shows homogeneous clusters of regions with the assessment of the graduates' theoretical readiness (left) and the assessment of the graduates' practical readiness (right). Regarding the assessment of the graduates' theoretical readiness, it is obvious that there are more clusters that are slightly linked, but there are two basic ones that are mutually more heterogeneous. The first cluster is represented by Prague, Central Bohemia, Moravia-Silesia, the Southwest, the Northwest, and Central Moravia. According to the K-means method, these regions show a slightly higher response rate for the graduates' assessment as "excellent", and a lower rate for other types of assessment when compared to the second cluster. The second cluster is represented by the Northwest, the Northeast, and the Southeast, and according to the K-mean method, these regions show significantly higher values for the rating "good", and slightly higher values for the rating "very good".

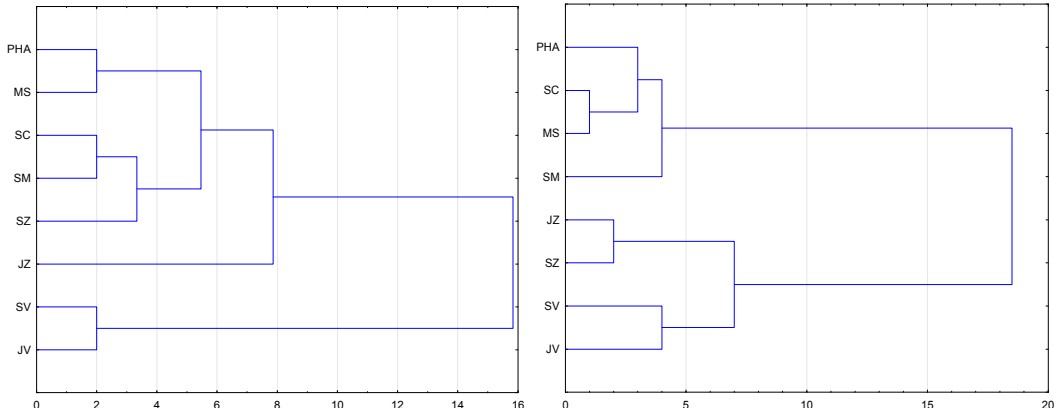

**Figure 8.** Hierarchical clustering, assessment of the graduates' theoretical readiness (**left**) and assessment of the graduates' practical readiness (**right**) for the midwife profession.

Regarding the graduates' practical readiness, it is clear that clusters formed more easily (the responses in the regions are closer to each other), and that there are two basic mutually more heterogeneous clusters. The first cluster is represented by Prague, Central Bohemia, Moravia-Silesia, and Central Moravia. According to the K-means method, these regions show a significantly lower response rate for the rating "very good" compared to the other cluster. The second cluster is represented by the Southwest, the Northwest, the Northeast, and the Southeast, and these regions rank the graduates' practical readiness significantly more often as "good" and "insufficient".

Figure 9 shows homogeneous clusters of regions that assess the change in the overall knowledge of graduates over time. Respondents commented on how the level of knowledge has changed (in the last 5 years) and rated this change using a point scale (significantly deteriorated, slightly deteriorated, approximately the same, slightly better, much better). From the dendrogram it is clear that three basic clusters, in which the regions are close to each other, have been formed. The first cluster consists of Prague and Central Moravia, where the most frequent response was that this level has slightly deteriorated; the second most frequent response was that it has deteriorated significantly. The second cluster consists of Central Bohemia, the Southwest, the Northwest, and Moravia-Silesia, with a medium level assessment and dominant responses that the knowledge level is about the same over time. The third cluster consists of the Northeast and the Southeast, which mainly evaluated the development of the knowledge level as slightly deteriorating.

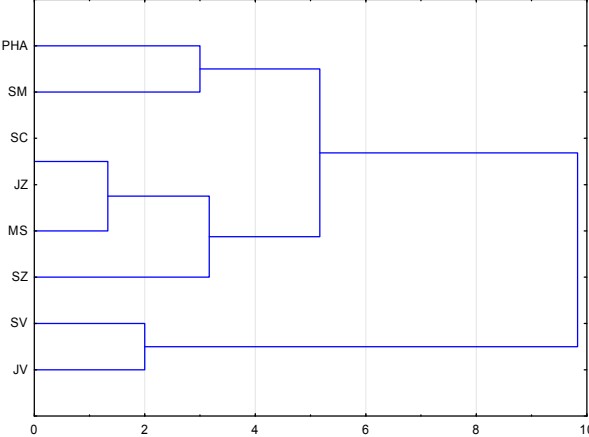

**Figure 9.** Hierarchical clustering, changes in the overall knowledge level of graduates in the midwife profession over the past 5 years.

Figure 10 (left) shows homogeneous clusters of regions that have most frequently reported lacking competencies in graduates in the relevant position. Respondents have been instructed to identify one knowledge, ability, or skill that is most often lacking in hired graduates. The competencies included language skills, teamwork in the relevant department, professional expertise for performance of the profession according to Regulation No. 55/2011 Coll. (newly No. 2/2016), computer skills, skills for communicating with patients, and time-management ability. The dendrogram created two major clusters. The first cluster consists of Prague, the Southwest, and Moravia-Silesia. The second cluster consists of Central Bohemia, the Northwest, the Northeast, Central Moravia, and the Southeast. The regions in the second cluster, more than the regions in the first, report that their employees primarily lack skills in communicating with patients and time-management ability. Both clusters congruently indicate the lack of professional expertise for performance of the profession according to Regulation No. 55/2011 Coll. (newly No. 2/2016) and teamwork in the relevant department. The lack of language skills or computer skills is only marginal. Healthcare facilities use various forms of continuous informal education of their employees in the relevant position, by which they compensate for the lack of knowledge and skills, or refresh existing ones. The most widely used forms of education in all regions are external specialized seminars and internal educational events; the least used form is master's-level study, and conferences and congresses are only used to a limited extent. Figure 10 (right) is a dendrogram of clusters, in which the capital city of Prague has a unique position, as it is the only cluster where master's-level study is the most frequently used form of employee education, followed by specialized conferences and congresses. Another cluster consists of Central Bohemia, the Southwest, Central Moravia, and Moravia-Silesia; these regions focus on education through specialized conferences and congresses, mostly through external specialized seminars. The third cluster consists of the Northwest, the Southeast, and the Northeast, where healthcare facilities most often focus on internal educational events, seminars, and also external specialized seminars.

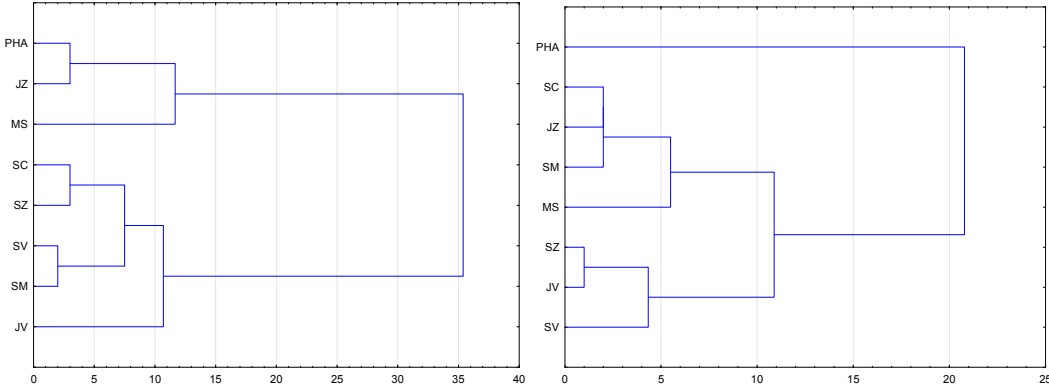

**Figure 10.** Hierarchical clustering, the most frequently lacking competencies in graduates in the midwife profession (**left**), Hierarchical clustering, educational activities most frequently used in the midwife profession (**right**).

## 5. Concluding Remarks

The results of the quantitative research show that the offer of workers in the midwife profession on the labor market is slightly insufficient in relation to the demand. The majority of addressed healthcare facilities stated that they are understaffed or slightly understaffed. Also, expectations for the future are pessimistic, and the prevailing opinion is that the coverage of this profession by skilled human resources is disproportionate, especially in Prague, Central Bohemia, Moravia-Silesia, and the Southwest regions. The results of structured interviews also show that the lack of midwives is manifested towards patients by the total or partial absence of the necessary supervision, care, and counselling of women during pregnancy. In practice, these activities are often provided by so-called *doulas*. However, they do not require the

same education as midwives. The research suggests that the overall educational level of graduates has improved over time. Central Bohemia, the Northwest, the Northeast, Central Moravia, and the Southeast regions agree that the greatest deficiencies in the competencies of new graduates are seen in the area of expertise according to Regulation No. 55/2011 Coll. (newly No. 2/2016), and that what graduates lack the most are the skills for communicating with patients. Conversely, current graduates are best equipped with language skills, computer skills, and time-management ability. The research also shows that overall practical readiness lags far behind theoretical readiness. The majority of regions (Prague, Central Bohemia, Moravia-Silesia, the Southwest, the Northwest, and Central Moravia regions) reported excellent or good theoretical readiness, but worse or slightly worse practical readiness. The forecast for the future is positive for practical readiness, as from this year the amendment to Act No. 55/2011 (newly No. 2/2016) significantly increases practical training in healthcare facilities (e.g., personally attending to at least 40 physiological deliveries during all stages of labor, including indicated episiotomy, post-delivery treatment and the examination of at least 100 mothers and babies in the early postpartum period, etc.). With regard to the fact that such significant legislative changes in the area of healthcare professionals' education took place in 2017, further analysis of the extent to which managers of departments of gynecology and obstetrics agree with the abolition of the credit system has been conducted. The survey results represent a beneficial basis for planning the education platform. They show that, in addition to the hard competencies given by the decree, attention should also be directed to soft competencies. The Study by Butler et al. (2018) confirms that the key competencies which have to be developed are communication skills and reasonable degrees of self-sufficiency, the ability use up-to-date knowledge in practice, and self and professional awareness. A further development of this topic can undertaken, for example, into a thematic analysis with a focus on identifying key events of the working lives of newly qualified midwives, as well as a study (Skirton et al. 2018) focusing on monitoring (1) the impact of the event on confidence, (2) gaps in knowledge or experience, and (3) articulated frustration, conflict, or distress. The results show that the professional public is divided on this issue, with a slight majority accepting this change. Despite the abolition of the credit system, the healthcare facilities finance lifelong learning for the midwife profession, with external educational activities prevailing slightly over internal activities in all monitored regions. The least used form is the master's degree study programme, which has a unique position in Prague. The regions of Central Bohemia, the Southwest, Central Moravia, and Moravia-Silesia focus on education through specialized conferences and congresses, mostly through external specialized seminars. Conversely, internal educational events in the form of seminars are more typical for the Northwest, the Southeast, and the Northeast regions. The midwives' initiative to improve their level knowledge and skills is only partly supported by the flexible component of their salaries or wages; there is even a large number of healthcare facilities that do not financially motivate their employees to conduct such activities.

**Author Contributions:** The authors contributed equally to this work.

**Acknowledgments:** The paper was prepared with the support of the Czech University of Life Sciences in Prague (Project IGA PEF No. 20171029). The authors of the article would like to thank the head nurses and the management of health facilities for the questionnaire survey, and also IGA for providing funding for the research.

**Conflicts of Interest:** The authors declare no conflicts of interest.

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
