# Peer review of "The Assessment of the Quality of Human Resources in the Midwife Profession in the Healthcare Sector of the Czech Republic"

_economies, doi:10.3390/economies6030038_

Round 1

Reviewer 1 Report

1) Introduction:

- In-depth litterature review is missing.

- There is no information about any research concerning midwife profession already performed in Czech Republic (or other countries in Europe), and especially about this kind of research related to the quality of human resources in healthcare sector.

2) The Models and Methods:

- Information about the number of private healthcare facilities and the number of public healthcare facilities would be interesting (there is only the total number of all facilities - 95) as well as the exact number of questionnaires returned from these two types of healthcare facilities.

- Description (at least short) of the questionnaire is missing (number of questions, etc.).

- What was the key of selection of the research sample - these 95 facilities? (specialist healthcare facilities (only gynaecology and obstetrics) or mutli-specialist healtcare facilities with the ginaecological/obstetric ward; their size, etc.)

- 68 questonnaires are relevant. What about 3 left? (71 returned in total) Why these 3 are excluded?

3) Data:

- Figure 1 - understaffing and overstaffing have the same color in the legend, hence the figure is not evident

- Figure 2 and 3 - do the evaluation vary depending on the type of facilities - public/private?

- Figure 4 - significant improvement and significant deterioration have the same color in the legend, hence the figure is not evident

- lines 219-220: "How do you rate the change in the overall level of knowledge of graduates in the midwife profession over the lat 5 years?" - Why do this question concern only 5 years and not 10? In the introduction it is said that the research focuses on the last 10 years. Moreover, it is the only question which refers to the time interval.

To be improved:

- more practical approaches not only counting and describing the numbers - perhaps some qualitative ideas/answers/impressioins/opinions...

- more recent published articles and reaserch

Author Response

Dear reviewers and respected editors of Economies,

 Thank you very much for the comments submitted in the review process of the study „The Assessment of The Human Resources Quality in the Midwife Profession in Healthcare Sector in the Czech Republic“. We have included your requirements into our article, these are red coloured (for better orientation). The chapter of Introduction was completed by domestic and international empirical studies in the field, usually qualitative research into the key competencies of midwives in international context. The Models and Methods were supplemented with more detailed information on the quantitative primary research conducted in healthcare facilities in the Czech Republic, focusing on the quality and quantity of workforce in the midwife profession. The chapter Discussion further outlines the possible directions of the examined issues development.

Looking forwards for future cooperation and best regards,

Authors

Reviewer 2 Report

The study is interesting. The topic the authors have selected is relevant because contents and value of jobs are increasingly changing over time and new skills are always necessary to acquired and developed. The study is mainly descriptive and elucidates the features of the observed phenomenon. Research question and theoretical background should be more clearly presented and questioned. The research is well presented. Thereby, it is necessary contextualize the research in comparison to and looking at an international and/or European situation and scenario. In the introduction, it is possible and necessary to enrich both bibliographical references in the text and add them in the references. It is necessary to precise, in the introduction, the aim of the study and the purpose the authors propose to achieve investigating the subject. The authors could provide further comments in the part related to discussion and conclusions. Theoretical, managerial and policy implications should be more elucidated in details by providing comments with regards to human resource management and organizational aspects.

Author Response

(The authors gave the same response as above.)
